# The Impact of Social Determinants of Health on Meningococcal Vaccination Awareness, Delivery, and Coverage in Adolescents and Young Adults in the United States: A Systematic Review

**DOI:** 10.3390/vaccines11020256

**Published:** 2023-01-24

**Authors:** Catherine Masaquel, Katharina Schley, Kelly Wright, Josephine Mauskopf, Ronika Alexander Parrish, Jessica Vespa Presa, Dial Hewlett

**Affiliations:** 1RTI Health Solutions, 3005 Boardwalk Dr # 105, Ann Arbor, MI 48108, USA; 2Pfizer Pharma GmbH, Linkstraße 10, 10785 Berlin, Germany; 3RTI Health Solutions, 3040 E Cornwallis Rd, Durham, NC 27709, USA; 4Pfizer, 235 E 42nd St, New York, NY 10017, USA; 5Pfizer, 500 Arcola Rd, Collegeville, PA 19426, USA; 6Medical Westchester County Department of Health, 134 Court St, White Plains, New York, NY 10601, USA; 7Infectious Disease Consultant Calvary Hospital Bronx, New York, NY 10461, USA; 8Microbiology & Immunology New York Medical College, New York, NY 10595, USA

**Keywords:** meningococcal disease, vaccine recommendation, meningococcal vaccination, health equity, health disparity

## Abstract

Vaccines remain a fundamental intervention for preventing illness and death. In the United States, suboptimal vaccine uptake in adolescents and young adults has been observed for meningococcal conjugate (MenACWY) and serogroup B meningococcal (MenB) vaccines, particularly among marginalized communities, despite current recommendations by the Advisory Committee on Immunization Practices. A systematic literature search was conducted in the MEDLINE and MEDLINE In-Process, Embase, Cochrane, PsychInfo, and CINAHL databases to identify both drivers of, and barriers to, MenACWY and MenB vaccine uptake in adolescents and young adults. A total of 34 of 46 eligible studies that presented outcomes stratified by race/ethnicity, geography, and socioeconomic status were selected for review. Results showed MenACWY and MenB vaccination coverage in adolescents and young adults is impacted by racial/ethnic, socioeconomic, and geographic disparities. Gaps also exist in insurance for, or access to, these vaccines in adolescents and young adults. Moreover, there was variability in the understanding and implementation of the shared decision-making recommendations for the MenB vaccine. Disease awareness campaigns, increased clarity in accessing all meningococcal vaccines, and further research on the relationships between measures of marginalization and its impact on vaccine coverage in adolescents and young adults are needed to reduce the incidence of severe infections.

## 1. Introduction

Invasive meningococcal disease (IMD) is a serious, unpredictable, life-threatening infection caused by the bacterium *Neisseria meningitidis* [1]. While relatively uncommon, it is a potentially fatal infection (case fatality rates of approximately 10% to 15% in the United States [US]) that can also cause long-term, disabling sequelae (e.g., neurological deficits, mobility, hearing deficits, and limb amputation) in 10% to 20% of survivors [1,2,3]. In the US, the highest rates of meningococcal disease are in infants, with a second peak in adolescence and early adulthood [1,4]. There are at least 12 serogroups that cause IMD, categorized by the polysaccharide capsules; however, 5 serogroups (A, B, C, W, and Y) cause more than 90% of the cases of IMD in the US [1]. Following the introduction of a vaccine targeting serogroups A, C, W, and Y in the US for adolescents in 2005, there was a decline in the incidence of meningococcal disease caused by these serogroups [1]. Since the introduction of the quadrivalent (serogroups A, C, W, and Y) meningococcal conjugate (MenACWY) vaccines, meningococcal serogroup B has emerged as the leading cause of IMD among US adolescents and young adults (16- to 23-year-old individuals), accounting for 61.7% (21 of 34 total cases) in 2018 and 48.8% (21 of 43 cases) in 2019 [1,3,5].

There are two types of meningococcal vaccines licensed and approved by the US Food and Drug Administration, and are recommended for use in healthy adolescents and young adults by the Centers for Disease Control and Prevention (CDC) Advisory Committee on Immunization Practices (ACIP): MenACWY vaccine (three products available) and serogroup B meningococcal (MenB) vaccine (two products available) [6]. The immunogenicity, effectiveness, and safety studies supporting the use in adolescents and young adults can be found in the prescribing information for Menactra (MenACWY-D), Menveo (MenACWY-CRM), MenQuadfi (MenACWY-TT), Trumenba (MenB-FHbp), and Bexsero (MenB-4C) [7,8,9,10,11]. The MenACWY vaccine currently has an ACIP “routine recommendation” for all healthy adolescents, with one dose administered at age 11–12 years and a booster dose administered at age 16 years [12]. The MenACWY vaccine is also recommended for infants, children, and all other age groups in individuals who are at high risk of meningococcal disease due to underlying health conditions or high risk of exposure [1,6]. The MenB vaccine is currently recommended by the ACIP for all children over the age of 10 years who are at high risk of meningococcal disease and is recommended only on the basis of “shared clinical decision-making” for healthy adolescents and young adults aged 16–23 years, with a two-dose series and a preferred age of vaccination between 16 and 18 years [12]. Shared clinical decision-making is “an approach where clinicians and patients share the best available evidence when faced with the task of making decisions, and where patients are supported to consider options, to achieve informed preferences” [13].

Using National Interview Survey-Teen (NIS-Teen) data, the CDC estimates the annual uptake of the MenACWY and MenB vaccines. In 2021 MenACWY vaccine uptake was 89.0% for individuals taking at least one dose of MenACWY vaccine, with the first dose recommended at age 11–12 years plus a booster dose recommended at age 16 years, but only 60.0% for those taking at least two doses of MenACWY or unknown type of meningococcal vaccine [14]. For the MenB vaccine, the uptake for one or more doses of the primary series was 31.4% through age 17 years—the upper age limit for the NIS-Teen [14]. This is likely an underestimate since some individuals may not be vaccinated until 18 years or older, frequently the time of entry for those attending college.

Several publications have suggested that the suboptimal uptake of the two types of meningococcal vaccines available in the US (MenACWY and MenB) may be related to inequities or disparities in awareness or access to these vaccines in adolescents and young adults of different racial or ethnic groups, socioeconomic groups, in different geographic areas in the US [15,16,17,18,19,20]. For example, Pruitt et al. [20], using the NIS-Teen data showed disparities in the uptake of the second dose of MenACWY vaccine by race/ethnicity and insurance status.

The objective of this systematic literature review (SLR) was to identify and summarize study findings on health disparities in awareness and uptake of meningococcal vaccines (MenACWY and MenB) using the Cochrane Equity Methods Group as a guide to identify PROGRESS-Plus factors associated with health inequities in US adolescents and young adults (10–25 years old; this age range was selected on the basis of approved labeling by the US Food and Drug Administration for MenACWY and MenB vaccines) [7,8,9,10,11,21]. PROGRESS-Plus is an acronym for the characteristics identified by Cochrane as being associated with health equity and the following factors: race/ethnicity, socioeconomic status, place of residence (urban/rural), occupation, gender, religion, education, social capital, personal characteristics associated with discrimination, features of relationships, and time-dependent relationships [21].

## 2. Methods

### 2.1. Search Strategy

Systematic literature searches were conducted from 1946 (database inception) to 31 May 2022 or 10 June 2022, depending on database, using pre-specified, reproducible PICOS (PICOS: Population and disease condition, interventions, comparators, outcomes, and study) criteria and PRISMA (PRISMA: Preferred Reporting Items for Systematic Reviews and Meta-Analyses) guidelines to identify studies reporting health inequities in rates of meningococcal vaccination. The electronic literature searches were conducted on 10 June 2022 in the following databases: MEDLINE and MEDLINE In-Process (using Ovid Platform; includes Daily Update), Embase (using Elsevier Platform), Cochrane database, PsychInfo (American Psychological Association database), and CINAHL (Current Index to Nursing and Allied Health Literature). The Embase electronic database search was conducted on 10 June 2022 and included the following annual conferences of interest: Infectious Disease Week (2020 and 2021), ISPOR (2020 and 2021), ISPOR Europe (2020–2021), the Academy of Managed Care and Specialty Pharmacy (AMCP; 2020–2022), and AMCP Nexus (2020–2021). Search terms included those related to the disease (e.g., meningococcal infections, *Neisseria meningitidis*), those related to meningococcal vaccination, and search terms related to health equity (e.g., health equity, healthcare disparities). A copy of the individual electronic database searches is provided in Appendix A). The electronic database searches included the following limitations: humans; English language; no comments, letters, or editorials; and US only. During screening, it was determined that studies published prior to 2012 would not be relevant, as the first MenB vaccine was not yet approved in the US. The first MenB vaccine was approved in 2014 [22]. Therefore, the protocol was amended to limiting inclusion of publications in the previous 10 years, thus allowing for more current identification of health inequities related to both MenACWY and MenB vaccination in individuals 10 to 25 years of age or individuals at increased risk of meningococcal infection in the US as stated in the research objectives for this SLR. Full details of our search strategy are in Appendix A.

### 2.2. Screening Strategy

Two reviewers independently screened titles to determine eligibility based on predefined PICOS criteria (Appendix A). Articles selected for full-text review (level 2 screening) were also screened by two independent reviewers to determine study eligibility based on the same predefined inclusion and exclusion criteria. If there was disagreement about study relevance, consensus was reached with a third researcher for both level 1 and 2 screenings.

### 2.3. Data Extraction

Data extraction included complete publication citation, study design and methods (i.e., sample size, data sources, funding source, duration of follow-up, and inclusion/exclusion criteria), patient demographics, (i.e., sample size, age, gender, ethnicity, place of residence (geography), and education and socioeconomic status), and meningococcal vaccine type and vaccine uptake.

Both qualitative and quantitative results were collected for factors impacting meningococcal vaccination knowledge/awareness, delivery, and coverage in individuals 10–25 years old in the US.

### 2.4. Quality Assessment

The Critical Appraisal Skills Programme (CASP) Quality Assessment Tools were used to evaluate study quality and the risk of bias for each publication included in this review. The CASP Quality Assessment Tools include checklists for the various study designs: case–control studies, cohort studies, economic evaluations, and qualitative studies [23,24,25,26]. Studies in our review were appraised by answering questions from three sections of the appraisal tool designed to determine (1) Are the results of the study valid? (2) Are the results clearly presented? and (3) Will the results help locally? To each question in the three sections, one of the following answers was recorded: “yes,” “no,” and “cannot tell.” The CASP Checklist was designed to be used as an educational, pedagogic tool, and a scoring system was not suggested [24,26,27,28]. Two researchers independently evaluated each study using the appropriate appraisal tool based on study design.

### 2.5. Analysis

The findings were descriptively summarized. Studies were grouped by the following factors and their impact on MenACWY and MenB vaccine coverage: race/ethnicity, geography, socioeconomic status, healthcare provider (HCP) and patient/guardian awareness of MenACWY and MenB disease and vaccines, health insurance, and access to healthcare. Given the heterogeneity of the studies included, no meta-analysis was performed.

## 3. Results

### 3.1. Search Results

A total of 2519, titles and abstracts were identified. Following removal of duplicates, 1577 unique titles and abstracts were screened (level 1 screening) by two independent reviewers to determine study eligibility based on the predefined inclusion and exclusion criteria. After reviewing the titles and abstracts (level 1 screening), 1411 articles were excluded; thus, 166 articles (163 full text + 3 Embase abstracts) were selected for full-text review (level 2 screening). Following level 2 screening, a total of 34 relevant sources were identified for inclusion (32 from electronic searches and 2 publications identified after a search of bibliographies; see PRISMA (Preferred Reporting Items for Systematic Reviews and Meta-Analyses) diagram in Figure 1).

### 3.2. Description of Included Studies

A total of thirty-four studies with quantitative or qualitative information were selected for inclusion. These studies focused on the following 3 PROGRESS-Plus factors: race/ethnicity, socioeconomic status, and geography. A brief summary of the included studies is provided in Appendix A.

### 3.3. Quality Assessment

The data quality for each study was assessed using the CASP qualitative or cohort study checklist and a list of questions addressing study validity, reported results, and the application of results locally. Appendix A present the results of the CASP study quality assessment for the included studies. Studies were characterized by responding with a “yes,” “can’t tell,” or “no.” For the qualitative study appraisal tool, questions 1 to 6 appraised whether the results of the study were valid, followed by questions 7–9, which examined whether the process for obtaining the results included consideration of ethical issues, if the data analysis was sufficiently rigorous, and if there was a clear statement of findings. Question 10 inquired about the extent to which the research is valuable. Similarly, for the cohort study appraisal tool, questions 1–6 were posed to appraise study validity, followed by questions 7–9, which asked the assessor to provide the quantitative measure reported and determine the level of precision of the results and respond whether the assessor believed the results. Questions 10–12 inquired whether study results will help locally. All studies were valid based on the response “yes” to the questions that addressed study validity and the “yes” response to questions related to results. Based on the reviewers’ responses, most of the studies could not be applied locally or were not generalizable due to the population studied.

### 3.4. Race/Ethnicity, Geography, and Socioeconomic Status and MenACWY and MenB Vaccination Coverage

#### 3.4.1. Race/Ethnicity and MenACWY and MenB Vaccine Coverage

Results from this review showed variations by race/ethnicity in MenACWY [18,20,29,30,31] and MenB [32,33,34,35] vaccine coverage (Appendix A).

A total of five studies presented estimates of differences in coverage of MenACWY vaccine between non-Hispanic participants described as Black or White adolescents or young adults [18,20,29,30,31]. Phillips et al. [30] conducted a longitudinal cohort study of 486 young men (ages 16–29 years) who had sex with men following an outbreak of meningococcal disease in Chicago. The median age of individuals enrolled in this study was 21.5 years, and nearly equal proportions of participants identified as Black (30.9%), Latino (30.2%), and White (25.2%). Of note, 19.6% of the individuals included in this study were positive for human immunodeficiency virus. The authors found that coverage of vaccination was higher in White persons than in Black persons, with an unadjusted odds ratio (OR) of 2.02 (95% confidence interval (CI), 1.20–3.38) [30]. The Pruitt et al. [20] study using NIS-Teen survey data from 2015 to 2017 (N = 63,299) indicated a slightly higher weighted percentage rate of coverage of at least one dose of MenACWY vaccine for Black (84.2%) and White (81.7%) adolescents aged 13 through 17 years. Similar results were found in another NIS-Teen (HCP reported) survey of data from 2017 (N = 3807), where Niccolai et al. [29] found that an increased number of Black adolescents aged 17 years were up to date with their MenACWY vaccine (defined as either receiving two doses of the MenACWY vaccine by age 17 years or one dose at age 16 or 17 years) compared with White adolescents (unadjusted OR = 1.60 (95% CI, 1.15–2.22) and adjusted OR = 1.81 (95% CI, 1.26–2.60)) when results were adjusted for sociodemographic (i.e., sex, race/ethnicity, poverty level, language of the interview, maternal age, maternal education, maternal marital status, and census region) as well as healthcare characteristics (type of health insurance, continuous health insurance since age 11 years, healthcare facility type, checkup at age 16 or 17 years, number of healthcare visits in past 12 months, and presence of a recommendation for a human papillomavirus (HPV) vaccination received by the adolescent (a proxy measure for providers’ practices related to vaccines; this question was not asked for other vaccines)). Coverage was also higher for Hispanic adolescents than for White adolescents (unadjusted OR = 1.11 (95% CI, 0.78–1.60) and adjusted OR = 1.63 (95% CI, 1.11–2.39)) [29]. Additional coverage data by race/ethnicity for the MenACWY vaccine were presented by Coyne-Beasley et al. [18]. Using 2008 North Carolina statewide Child Health Assessment and Monitoring Program surveys and the North Carolina Behavioral Risk Factor Surveillance System, the investigators analyzed data provided by 1281 caregivers (92% of caregivers were parents) of children aged 11–17 years. Bivariate and multivariate analyses showed that for those parents aware of the MenACWY vaccine before the survey, non-Hispanic African American children were more likely to have received the MenACWY vaccine compared with non-Hispanic White adolescents with adjusted coverage of Black versus White adolescents (bivariate OR = 2.19 (95% CI, 1.32–3.63) and multivariate OR = 2.17 (95% CI, 1.29–3.65)) [18]. Additionally, coverage rates for the MenACWY vaccine found by Kurosky et al. [31] using claims data from Medicaid patients during 2011–2016 were higher for Black adolescents (N = 189,879) versus White adolescents (N = 227,156; adjusted OR = 1.33; 95% CI, 1.31–1.35).

A total of four studies have estimated differences in coverage of the MenB vaccine between non-Hispanic Black and White adolescents [32,33,34,35]. La et al. [32] utilized a cross-sectional, retrospective, exploratory analysis of preexisting 2017–2018 NIS-Teen survey data on 7288 US individuals aged 17 years. The authors estimated unadjusted values for overall percent coverage of 16.1% for non-Hispanic Black persons and 13.5% for non-Hispanic White persons for ≥one dose of MenB vaccine. However, when the data were controlled for demographic variables, insurance status, and previous vaccines, the adjusted OR was 0.74 (95% CI, 0.48–1.14), indicating that this difference was not statistically significant. The other three studies found lower MenB coverage rates for Black adolescents compared with White and other/unknown (including White) adolescents [33,34,35]. Bart et al. [34] conducted a cross-sectional study analyzing data from 16- to 18-year-olds with a record in the Philadelphia immunization registry (KIDS Plus II) to identify sociodemographic factors associated with MenB vaccine coverage. A total of 85,489 individuals aged 16–18 years were identified, of which 54.5% were Black. Multivariate logistic regression analysis showed significant variation in the proportion of participants receiving ≥one dose of MenB, with individuals who reported unknown or other race more likely to have received MenB compared with Black/African American individuals (adjusted OR = 1.36 and 1.24, respectively; *p* < 0.0001) [34]. In a 2022 study of MenB vaccine series completion rates in 16- to 23-year-olds in the MarketScan Medicaid database who had taken the first MenB vaccine, Packnett et al. [33] estimated an overall series completion rate of 44.7% among the 57,082 individuals in the Medicaid database and a completion rate of 56.7% among the 156,080 individuals in the commercial data set. Completion rates were also lower among Black persons compared with White persons (Medicaid only), with an adjusted relative risk of 0.86 (95% CI, 0.84–0.88) [33]. Watkins and Feemster [35] observed a similar trend following a retrospective cohort analysis of 45,428 16- to 23-year-olds from 31 primary care sites in a pediatric care network (23 October 2015 through 30 April 2017). In this population, a total of 9393 adolescents received ≥one dose of MenB. Of those, a higher proportion who were White (27%) or Asian (22%) received ≥one dose of MenB compared with those who were Black (18%) [35].

#### 3.4.2. Geographical Factors and MenACWY and MenB Vaccine Coverage

Disparities in MenACWY and MenB vaccine coverage varied by geographic census region and by population density. A total of two studies presented estimates of differences in coverage of the MenACWY vaccine by geographic region within the US [29,31]. Using 2017 NIS-Teen survey data, Niccolai et al. [29] estimated the percentage of adolescents who were up to date with two doses of MenACWY vaccine. These estimates indicated that coverage of the MenACWY vaccine was most complete in the Northeast census region, with adjusted ORs for the other regions relative to Northeast of 0.72 (95% CI, 0.52–0.99) (Midwest), 0.36 (95% CI, 0.27–0.49) (South), and 0.35 (95% CI, 0.28–0.54) (West). Kurosky et al. [31] conducted a retrospective analysis of deidentified patient-level healthcare claims in the Commercial Claims and Encounters (CCAE) database and multistate Medicaid MarketScan Research Databases. The CCAE database included data from nearly 30 million children as of 2015. The authors also estimated coverage of the MenACWY vaccine by census regions. They estimated coverage for the first dose in those aged 10.5–13 years ranged from 67.33% in the West region to 71.96% in the Northeast region and 73.73% in the North Central region. A lower coverage rate and a greater disparity among regions in coverage for the second dose was shown in those aged 15.5 to 18 years, with coverage ranging from 39.56% in the West region to 59.38% in the Northeast region. In multivariate analyses, the individual likelihood of receiving the MenACWY vaccine was lowest in the Northeast compared with other regions (adjusted OR > 1 for the other regions) when controlling for patient characteristics and access variables in the analyses [31].

A total of two studies presented estimates of differences in coverage of MenB vaccine by region of residence [32,33]. La et al. [32] used 2017–2018 NIS-Teen survey data and determined that unadjusted coverage rates were highest in the Northeast region (18.3% for ≥one dose and 9.3% for ≥two doses) and lowest in the South region (14.6% for ≥one dose and 6.3% for ≥two doses); however, in all regions, estimated coverages were higher than that for New England in the multivariate analyses, with an adjusted OR of 1.90 (95% CI, 1.24–2.92) for the South Atlantic and 1.64 (95% CI, 1.03–2.62) for the Mountain region. In the study by Packnett et al. [33], regional data were available only for the portion of the population with commercial insurance. On the basis of that cohort, they estimated unadjusted series completion rates for MenB vaccine were highest in New England (66.4%) and lowest in the Mountain region (49.3%). This ordering was maintained for the adjusted relative risks, where all regions had adjusted relative risks of <1 compared with New England, with those for the Mountain region being the lowest at 0.75 (95% CI, 0.73–0.78) [33].

There were six studies that presented estimates of differences in coverage of the MenACWY or MenB vaccine by population density such as rural or urban [20,31,35,36,37] (Appendix A).

In the Kurosky et al. [31] study, coverage of the MenACWY vaccine was compared in rural and urban areas. In those aged 10.5–13 years, coverage of the first recommended dose was lower (58.21%) in rural regions compared with 73.69% in urban regions. A different coverage rate was also observed for the second dose among urban and rural regions in those aged 15.5 to 18 years, with a coverage of 34.24% in the rural regions and 51.10% in the urban regions, demonstrating a similar disparity to that of first-dose recipients. Similar results were obtained in the multivariate analysis, with an adjusted OR of 0.72 (98% CI, 0.71–0.73) for rural residents compared with urban residents [31]. The study by Gowda et al. [37] used data from the Michigan Care Improvement Registry (MCIR) for 2006–2010 to compare coverage of the MenACWY vaccine in residents in urban, suburban, or rural areas before a state school mandate was passed in 2010. This revealed a very small difference in the unadjusted coverage in the different regions (44.4–48.9%) and in the adjusted OR comparing suburban (0.91 (95% CI, 0.90–0.93)) and large-/small town with urban residents (0.95 (95% CI, 0.94–0.97)). Using data from NIS-Teen survey from 2015–2017, Pruitt et al. [20] examined patterns of coverage by rural or urban residence. They found that the MenACWY vaccine coverage for at least one dose was 82.8% overall; however, coverage in rural regions was 73.3% in contrast to 83.9% coverage in urban regions. A study by Bernstein et al. [36] using 2012 NIS-Teen survey data compared “red” states with “blue” states based on voting preferences (Republican versus Democratic, respectively) in the 2012 US Presidential election and showed that coverage of the MenACWY vaccine was higher in “blue” states (79.3%) than in “red” states (72.8%), with an adjusted difference of 14.1% (95% CI, 7.5–21.0%) when the results were controlled for sociodemographics (obtained from the US Census Bureau and included median household income, Gini index of income inequality, percentage of the population below the federal poverty level, percentage of the population with a bachelor’s degree or higher, and percentage of the population that was African American or Hispanic), access to care, and state vaccine policy. Finally, using 2020 NIS-Teen survey data, Pingali et al. [38] compared coverage of the first and second doses of the MenACWY vaccine for metropolitan statistical area (MSA) principal cities with that of non-MSA and MSA non-principal cities. For first doses, coverages were 85.7% (non-MSA), 89.4% (MSA non-principal city), and 90.2% (MSA principal city), while second-dose coverages were 50.1% (non-MSA), 58.5% (MSA non-principal cities), and 50.6% (MSA principal city) (Appendix A). Only the coverages for first dose in non-MSAs and second dose in MSA non-principal cities were statistically significantly different from those in the other regions. A study conducted by Watkins and Feemster [35] evaluated MenB vaccine coverage by pediatric care site (i.e., urban vs. suburban). In this study, the percentage of adolescents who received ≥one dose of MenB was similar (21% Urban and 20% suburban) [35].

#### 3.4.3. Socioeconomic Status and MenACWY and MenB Vaccine Coverage

Studies assessing socioeconomic status and MenACWY and MenB vaccine coverage have identified significant variation in potential sociodemographic disparities in MenACWY and MenB coverage (Appendix A) [20,29,32,34,38].

A total of three studies reported estimates of differences in coverage of MenACWY vaccines by socioeconomic status [20,29,38]. Niccolai et al. [29] estimated the percentage of adolescents who were up to date with two doses of the MenACWY vaccine using 2017 NIS-Teen survey data. The mean percentage for individuals up to date with MenACWY vaccination was 54.6% for those with annual incomes below the poverty level, 47.3% for those above the poverty level and earning ≤$75,000 annual income, and 50.7% for those above the poverty level and earning > $75,000 annual income. Compared with individuals below the poverty level, the unadjusted OR was 0.75 (95% CI, 0.52–1.08) for those above poverty levels with annual incomes ≤$75,000 and 0.86 (95% CI, 0.60–1.22) for those above poverty levels with annual incomes >$75,000. These differences were not statistically significant, and thus poverty level was not included in their multivariate analyses [29]. The Pingali et al. [38] study used 2020 NIS-Teen survey data to compare coverages of the first and second doses of the MenACWY vaccine for individuals with annual incomes below the poverty level compared with those with annual incomes at or above the poverty level. The rates of first doses were similar across different regional types (non-MSA, MSA non-principal city, MSA principal city) for those with annual incomes below the poverty level (86.1%, 87.2%, and 91.6%, respectively) compared with those at or above the poverty level (85.6%, 90.2%, and 89.4%, respectively). However, the rates of second doses were lower in those below the poverty level (47.4%, 47.6%, and 48.6%, respectively) than in those above the poverty level (50.2%, 61.2%, and 50.2%, respectively) [38]. Using 2015–2017 NIS-Teen survey data, Pruitt et al. [20] also indicated a similar rate of coverage of at least one dose of the MenACWY vaccine in adolescents up through age 17 years for those below the poverty level (83.7%), above the poverty level but ≤$75,000 annual income (80.4%), and above the poverty level but >$75,000 annual income (84.1%), but they did not estimate coverage of the second dose separately.

There were two studies that estimated the impact of the poverty level on MenB coverage [32,34]. Bart et al. [34] analyzed data from all 16- to 18-year-olds with a record in the Philadelphia immunization registry (KIDS Plus II) from 23 October 2015 through 31 July 2017. A total of 85,489 individuals aged 16–18 years were included in the multivariate regression analysis. A significant variation in the proportion of MenB recipients was identified and indicated that individuals living in a neighborhood with a median income of >$100,000 were more likely to receive the MenB vaccine compared with those residing in a neighborhood with a median income < $20,000 (adjusted OR = 1.63; *p* < 0.0001) [34]. La et al. [32] used 2017–2018 NIS-Teen survey data to estimate the percentage of individuals who received at least one dose of MenB vaccine among adolescents aged 17 years. They did not find a significant difference according to poverty level, with 19.6% (95% CI, 15.0–25.1%) for those below the poverty level, 14.1% (95% CI, 11.4–17.4%) for those above the poverty level but with annual income ≤$75,000, and 14.9% (95% CI, 12.9–17.1%) for those above the poverty level but with annual income > $75,000 [32].

### 3.5. The Impact of Health Insurance on MenACWY and MenB Vaccine Coverage

Health insurance coverage or lack of health insurance has an impact on the uptake and prescribing of vaccines, including the MenACWY and MenB vaccines [37,39,40,41,42]. A study by Singer et al. [39] examined the extent to which commercial insurance plans covered vaccination with MenACWY in adolescents. This study surveyed a national sample of private health insurance plans to determine coverage of recently recommended vaccines, of which the MenACWY vaccine was included, between December 2008 and June 2009, before passage of the Affordable Care Act (ACA) 2010. The researchers showed that health insurance coverage was incomplete, with for example, complete coverage of the MenACWY vaccine in only 66% of the plans. A total of four studies presented observed differences in coverage of the MenACWY vaccine in adolescents by insurance status before passage of the ACA [37,40,41,42]. In a study using 2009 NIS-Teen survey data, Gowda and Dempsey [40] performed bivariate and multivariate analyses of state-specific coverage rates among 13- to 17-year-olds of adolescent vaccines, including the MenACWY vaccine. They did not show higher coverage of the MenACWY vaccine in states with a higher percentage of adolescents enrolled in public insurance programs such as Medicaid and the Children’s Health Insurance Program (CHIP) or in states where the Medicaid reimbursement rates for vaccination were higher. In a retrospective database study from 2006 to 2010 in the MCIR, supplemented with Medicaid data where relevant, Gowda et al. [37] estimated the proportions of the 597,846 MenACWY vaccine doses given to adolescents aged between 11 and 17 years (statewide coverage rate of 46.5%) by insurance type: 35% for Medicaid or public insurance; 51%, private insurance; 3%, uninsured; 0%, both public and private insurance; and 11%, unknown. They did not include insurance status in the multivariate analyses of coverage. However, in a study that included three annual cohorts of 11- to 12-year-olds and that also used the MCIR in 2006–2008, Rees-Clayton et al. [41] assessed vaccine coverage for three adolescent vaccines, including the MenACWY vaccine, and showed higher coverage in adolescents who had ever been covered by Medicaid compared with those who had never been covered by Medicaid (for the MenACWY vaccine in 2008—34.8% versus 20.8%) with the odds of being vaccinated with the MenACWY vaccine being two times higher for those with Medicaid than that for those with no Medicaid (OR = 2.03; 95% CI, 2.00–2.07). Seib et al. [42] included a survey of 686 parents in Georgia among middle and high school students about adolescents’ receipt of the MenACWY vaccine and other adolescent vaccines as part of a controlled trial of parent education interventions during 2011–2013. In this study, in which 75% of enrollees were African American and 16% were Caucasian, 60% of those with Medicaid insurance, 59% of those with private insurance, and 45% of those with no insurance had received the MenACWY vaccine [42].

An additional three studies of the MenACWY or MenB vaccines estimated the difference in coverage rates by insurance status after passage and implementation of the ACA, which mandated coverage of ACIP-recommended vaccines for most insurance plans [33,43,44]. In a study using data from the NIS-Teen survey in 2015, Lu et al. [43] showed similar coverage rates for the MenACWY vaccine for adolescents with private insurance (81.7%; 95% CI, 80.2–83.0%) and those with Medicaid insurance (82.1%; 95% CI, 80.3–83.8%). A second study using data from the NIS-Teen survey (2017 and 2018) found differences in the association of health insurance status with MenB vaccine uptake for 17-year-old adolescents [44]. For adolescents who resided in a state with a meningococcal vaccine school requirement, health insurance was not significantly associated with MenB vaccine coverage. However, for adolescents who resided in a state with no meningococcal vaccine requirement, those with private health insurance were less likely to have received the MenB vaccination compared with those who had other types of insurance, including Medicaid or no insurance (adjusted OR = 0.40; 95% CI, 0.24–0.66) [44]. The study by Packnett et al. [33] of MenB vaccine series completion rates in 16- to 23-year-olds for those with at least one MenB vaccine comparing MarketScan Commercial and Medicaid databases estimated a MenB vaccine series completion rate of 56.7% for those in the commercial database and 44.7% for those in the Medicaid database, indicating a higher series completion rate for MenB vaccine for those with private health insurance than for those with Medicaid insurance [33].

### 3.6. The Impact of Healthcare Access on MenACWY and MenB Vaccine Coverage

The American Academy of Pediatrics (AAP) published the Child and Adolescent Immunization Schedule, which provides recommendations for children and adolescents aged 18 years or younger in the US [45]. The AAP recommends on-time immunization of all children and adolescents from infancy through adolescence during annual preventive care visits [45]. There are limited data on barriers to the MenACWY and MenB vaccines for rural adolescents, and more research is needed on barriers such as lack of HCPs, financial constraints, and transportation hindrances [31,46,47]. There were three studies presented with estimates of differences in uptake of the MenACWY or MenB vaccine by healthcare access [31,46,47]. Cataldi et al. [46] administered an internet and mail survey between June and August 2019 to practicing public health nurses, pediatricians, and family medicine clinicians in the state of Colorado. In this study, the authors assessed rural-urban differences in logistical barriers to adolescent vaccination (MenACWY, Tdap, HPV, and influenza). When HCPs were asked whether they agreed with the following statement, “Adolescents do not come to primary care for annual well visit,” the responses from urban and rural providers were as follows: strongly agree, 7% versus 15%; somewhat agree, 49% versus 47%; somewhat disagree, 32% versus 32%; and strongly disagree, 12% versus 5% (*p* < 0.01) [46]. In a retrospective analysis aimed at identifying factors associated with MenACWY vaccine uptake among adolescents, Kurosky et al. [31] found that, based on databases from 2011 to 2016, Medicaid patients’ uptake of ≥one MenACWY vaccine was significantly and consistently lower in older adolescents aged 15.5 through 18 years (48.9%; 205,131/419,814) than in younger adolescents aged 10.5 through 13 years (71.7%; 270,186/376,825) (*p* < 0.001). Older adolescents who did not receive the MenACWY vaccine had a mean (standard deviation (SD)) of 1.12 (1.37) preventive care/well-child visits (per individual) compared with 1.43 (1.98) visits in younger adolescents. After adjusting for individual demographic and healthcare resource utilization characteristics, the authors found that older adolescents had a significantly lower likelihood of receiving a MenACWY vaccine compared with younger adolescents, despite routine ACIP recommendations, complementing NIS-Teen survey data [31]. For the MenB vaccine, Ghaswalla et al. [47] conducted a retrospective analysis of pooled 2016–2018 NIS-Teen survey data (N = 10,995), including adolescents with adequate provider-reported vaccination data who were aged 17 years at the time of the survey. MenB vaccination coverage (≥one dose of MenB vaccine at any age) was estimated overall and by individual-level characteristics. The following factors were significantly associated with higher likelihood of receiving ≥one dose of MenB vaccine: (1) Medicaid versus private/other insurance, (2) individuals aged 16–17 years at last checkup, (3) receipt of an HCP recommendation for an HPV vaccine, (4) up-to-date status with HPV vaccination, (5) up-to-date status with MenACWY vaccination, and (6) residence in South Atlantic or Mountain census divisions versus New England [47].

### 3.7. Healthcare Professionals’ Knowledge and Awareness of ACIP Meningococcal Vaccine Routine and Shared Decision-Making Recommendations and Disparities in Recommendations for MenACWY and MenB Vaccines

Real-world studies have shown that the HCPs’ interpretation of current ACIP recommendations for routine MenACWY vaccination and shared clinical decision-making recommendations for MenB vaccine has led to disparities in implementation of routine discussions, and prescribing recommendations with patients and their caregivers [48,49]. These studies examined the degree to which pediatricians and family practitioners are knowledgeable and aware of the ACIP’s recommendations for MenB and MenACWY vaccines [49]. Huang et al. [49] conducted a web-based survey between August and October 2017 among HCPs recruited through a global panel of >55,600 US HCPs reporting with membership with the American Medical Association. Of the 529 respondents, 81.5% (431/529) reported that they prescribed MenB/MenACWY vaccines to their eligible adolescent/young adult patients, 0.9% (5/529) prescribed only MenB, and 17.6% (93/529) prescribed MenACWY vaccines only. When respondents were asked to indicate their interpretation of ACIP’s recommendations, results showed that only 7% of HCPs correctly interpreted shared clinical decision-making, and 77% of those surveyed indicated that they consistently interpreted the MenACWY vaccine routine recommendations. Factors such as age, ethnicity, student status, living arrangement, insurance coverage, whether the patient had received other vaccines in the past, or whether the patient was treated by a frequent MenB vaccine prescriber varied between patients who had received the MenB vaccine versus those who had received the MenACWY vaccine only [49]. When asked what factors affected the decision of the HCP to prescribe or not prescribe meningococcal vaccines, the HCPs most influential reason to prescribe were guideline considerations for the MenACWY vaccine and disease-related factors for MenB. For both vaccines, the reason ranked the highest for not prescribing was “financial considerations.” In fact, HCPs who prescribed both MenACWY and MenB vaccines had a higher percentage of privately insured patients than those who only administered MenACWY vaccines (61.19% (SD: 24.90%), *p* = 0.003 versus 52.58% (SD: 26.66%), *p* = 0.008; unadjusted) [49].

Results from a second survey of primary care provider (pediatricians and family physicians) knowledge regarding MenB vaccine conducted by Kempe et al. [48] observed similar discrepancies in knowledge and interpretation of the ACIP’s recommendations for meningococcal vaccination among the 660 respondents (response rate of 72% (660/916)). Of the 660 respondents, 374 were pediatricians and 286 were family physicians. Only 56% of pediatricians and 38% of family physicians were able to correctly interpret the shared decision-making recommendations for MenB. However, 80% of pediatricians and 85% of family practitioners surveyed were able to correctly interpret the routine recommendations for the MenACWY vaccine. Additionally, 55% of physicians surveyed did not know that private insurance would pay for vaccines based on shared decision-making, and 51% did not know that vaccines with a shared decision-making recommendation were covered by the Vaccine for Children program [48]. The most common misunderstanding among the physicians surveyed was thinking that a routine recommendation in a subgroup of patients was shared decision-making recommendation for MenB rather than part of the routine recommendations for the MenACWY vaccine. Kempe et al. [48] also noted that this lack of understanding might deter physicians from discussing the vaccine with their patients and families or providing the vaccine at all.

In a follow-up study conducted by Kempe et al. [50], the fact that the MenB vaccine was given a shared decision-making recommendation as opposed to a routine recommendation by the ACIP was found to be a major issue associated with not recommending MenB vaccination for both pediatricians and family physicians. Overall, when asked about current practices regarding MenB, 51% of pediatricians and 31% of family physicians reported “almost always/always” or “often” initiating a discussion during a routine visit for 16- to 18-year-olds and 60% of pediatricians and 40% of family physicians initiated the discussion during a pre-college physical examination. Additionally, physicians who reported “somewhat/not at all aware of MenB vaccine” (risk ratio = 0.32; 95% CI, 0.25–0.41) and those practicing in a health maintenance organization (risk ratio = 0.39; 95% CI, 0.18–0.87) were found to be less likely to initiate a discussion about the MenB vaccine [50].

### 3.8. Parent/Guardian Knowledge and Awareness of Meningococcal Disease and Meningococcal Vaccines and the Impact on MenACWY and MenB Vaccine Coverage

Several studies have shown that there is a lack of meningococcal disease and meningococcal vaccine knowledge among parents [12,18,51,52,53,54]. This lack of awareness has resulted in disparities in meningococcal vaccination rates among those in the 10- to 23-year age range. A total of six studies surveyed parent/guardian meningococcal disease and/or vaccine awareness (Appendix A) [12,18,51,52,53,54]. A total of three of these studies asked the parent/guardians to indicate whether they were aware of meningitis or meningococcal disease [12,51,52]. The proportion of parents/guardians who reported meningococcal disease awareness ranged from 50–96% [12,51,52]. All six of the surveys asked whether the parent/guardian was aware of meningococcal vaccines [12,18,51,52,53,54]. The proportion of parents/guardians who indicated they were aware ranged from 20% to 65% for the MenACWY vaccine [18,51,52,54] and from 20% to 43% for the MenB vaccine [12,51,53]. Disparities in awareness for the MenACWY and MenB vaccines were observed to be related to parental confusion, race/ethnicity, lack of HCP recommendation for one or both vaccines and socioeconomic status [12,18,51,52,53,54].

All six of these studies observed lack of awareness or confusion regarding the availability and coverage provided by each of the meningococcal vaccines (e.g., MenACWY and MenB) [12,18,51,52,53,54]. Richardson et al. [51] observed that parents were more likely to report having heard about the MenACWY vaccine (61%) than the MenB vaccine (40%). Results from this study also showed uncertainty and confusion among the parents surveyed about the MenB vaccine due to the existence of another meningitis vaccine and limited HCP recommendations. Basta et al. [12] surveyed 445 parents of teens attending high school in Minnesota (2017–2018). When parents were asked about meningococcal vaccines, 75.5% (95% CI, 71.2–79.4%) reported that they were aware of meningococcal vaccines in general, and 71.7% of the respondents considered themselves at least “somewhat knowledgeable” about meningococcal vaccines. However, the majority of parents had not heard of the newly introduced MenB vaccines Bexsero^®^ (80.0%; 95% CI, 76.0–83.6%) and Trumenba^®^ (82.0%; 95% CI, 78.1–85.5%) and approximately 68.8% of the parents, (95% CI, 64.2–73.0%) had not heard of MenACWY vaccines [12]. Coyne-Beasley et al. [18] examined MenACWY vaccine awareness among 1281 parents of adolescents aged 11 to 17 years. Overall, 65% of parents reported they had heard of meningococcal vaccine [18]. Bivariate analyses showed that parents were more likely to have heard of the MenACWY vaccine if their children were aged 16–17 years (compared with those aged 11–12 years) and had a primary care medical provider (bivariate OR = 1.97; 95% CI, 1.36–2.86; *p* < 0.001), or had received a preventative medical evaluation within the past year (bivariate OR = 1.65; 95% CI, 1.18–2.30; *p* < 0.05) [18]. An online survey of 619 adults with ≥one dependent aged 16–19 years was conducted by Srivastava et al. [53] to determine factors associated with MenB vaccine awareness and utilization. Survey participants were identified from the 2016 KnowledgePanel^®^ database. A total of 467 (43%; weighted percentage) participants were aware of MenB vaccines [53]. Greenfield et al. [54] conducted in-person surveys of Hispanic, Somali, and Ethiopian/Eritrean adolescents (n = 45) and parents of adolescents (n = 157) to assess knowledge of recommended adolescent vaccines, including the MenACWY vaccine, in diverse ethnic communities. Overall, 33% (52/157) of parents had heard of the MenACWY vaccine [54]. Painter et al. [52] conducted in-person interviews of 30 immigrant mothers of adolescent daughters to assess vaccine-related knowledge. A total of fifty percent (15/30) of mothers reported ever hearing of meningococcal disease and 20% (6/30) had heard of meningococcal vaccine [52].

While three of the six studies observed significant differences in parental/guardian vaccine knowledge or awareness by race/ethnic group, no consistent trends were observed [18,53,54]. In a multivariate analysis of parents who were aware of the MenB vaccine compared with those who were not, Srivastava et al. [53] showed that there was significantly more MenB vaccine awareness among parents/guardians of White, non-Hispanic adolescents versus those of Hispanic adolescents; Black, non-Hispanic adolescents; and adolescents of other race/ethnicity (OR = 2.20 [95% CI, 1.09–4.46]) (Appendix A) [53].

However, Coyne-Beasley et al. [18] observed that parents/guardians were less likely to have heard of meningococcal vaccine if their children were Hispanic compared with non-Hispanic White children (OR = 0.50; 95% CI, 0.28–0.89). Greenfield et al. [54] conducted in-person surveys of Hispanic, Somali, and Ethiopian/Eritrean adolescents (n = 45) and parents of adolescents (n = 157) to assess knowledge related to recommended adolescent vaccines, including the MenACWY vaccine. Overall, 33% (52/157) of parents had heard of the MenACWY vaccine, and among adolescents, 20% had heard of the MenACWY vaccine. Following a bivariate analysis of parent survey responses, there were significant differences in MenACWY vaccine awareness among the three ethnic groups (*p* < 0.001), with Hispanic parents having the greatest likelihood of having heard of any of the adolescent vaccines (Tdap, MenACWY, or HPV) and parents of Somali descent being least likely to have heard of any of the adolescent vaccines [54].

Disparities associated with lack of HCP recommendations were identified in five of the six studies [12,18,51,53,54]. Both Basta et al. [12] and Coyne-Beasley et al. [18] identified lack of a recommendation from an HCP as one of the most common reasons for not vaccinating their child with the proportion of parents reporting this ranging from 24.7–33.9%. Richardson et al. [51] estimated that only 31% of parents received a physician recommendation and of those who received a recommendation, approximately 9–22% had initiated the MenB vaccine. Additionally, under the assumption that none of the parents who were unaware of the MenB vaccine had received a HCP recommendation and had their adolescent vaccinated, the investigators estimated that up to 70–80% of 16- to 17-year-olds had missed opportunities to receive the MenB vaccine [51]. Of the 43% of parents surveyed by Srivastava et al. [53] who reported they were aware of the MenB vaccine, 69% indicated that they had learned about MenB vaccines from an HCP. Further multivariate analysis supported that vaccination/intent to vaccinate was significantly more likely if a provider had recommended the MenB vaccine (OR = 4.81; 95% CI, 2.46–9.35) [53]. In addition to in-person surveys of Hispanic, Somali, and Ethiopian/Eritrean parents of adolescents, Greenfield et al. [54] conducted three focus groups with mothers of 11- to 18-year-olds to assess knowledge, attitudes and barriers related to recommended adolescent vaccines. Most parents (92%) in the focus groups identified their doctor as a trusted source of health information. Among those who had not vaccinated their adolescent, 60% reported that they would have their adolescent vaccinated with Tdap or MenACWY if their doctor had recommended vaccination [54]. In addition to these studies, Kricorian et al. [55] conducted a health literacy survey among 43 adult females (Hispanic (n = 28) and non-Hispanic (n = 15)) attending a health fair in an underserved area of Los Angeles, California. Results showed a significantly lower percentage of Hispanic versus non-Hispanic women reporting recognition of the word “meningitis” (15% vs. 60%, *p* < 0.01) [55].

Disparities in parent/guardian awareness associated with socioeconomic status were observed in two studies [18,53]. Srivastava et al. [53] surveyed vaccine awareness based on average annual income. Results from a Classification and Regression Tree (CART) analysis indicated that some of the most influential variables associated with awareness of MenB vaccine included annual household income (relative importance, 14.1%) and property ownership (relative importance, 13.8%). Coyne-Beasley et al. [18] observed greater parental awareness of meningococcal vaccine prior to survey (n = 1281) among parents with household incomes ≥$50,000 (72.9%) compared with <$50,000 (57.8%) and bivariate analyses identified that parents were less likely to have heard of meningococcal vaccine if their annual household income was <$50,000 or not reported (*p* < 0.05).

There were four studies that provided vaccination coverage results based on meningococcal disease and vaccine awareness [18,51,52,54]. Of the 70 parents in the Richardson et al. [51] study who reported that they had heard of MenB vaccine, 50% reported that their 16- to 17-year-olds had received at least one dose of the MenB vaccine. However, when the investigators obtained records from the state vaccine registry, only 23% (16/70) had actually received at least one dose of MenB vaccine. Coyne-Beasley et al. [18] reported that among surveyed parents who were aware of the meningococcal vaccine and provided vaccination status, 44% (280/703) had received the MenACWY vaccine. Greenfield et al. [54] reported that of the respondents from all three ethnic groups studied (Hispanic, Somali, and Ethiopian/Eritrean) who had heard of Tdap or MenACWY vaccines, 77% reported that their adolescents had received these vaccines. Among the 30 low-income, uninsured, predominantly Latin American immigrant mothers of adolescent daughters who participated in the survey conducted by Painter et al. [52], 70% (21/30) of mothers either did not know whether their daughter had received the MenACWY vaccine or had never heard of the MenACWY vaccine. Only 16.7% (5/30) had reportedly received the MenACWY vaccine.

### 3.9. Lack of Healthcare Professional Recommendations for Meningococcal Vaccination and the Impact on MenACWY and MenB Vaccine Coverage

One of the of the challenges impacting vaccination coverage is HCP’s willingness to recommend vaccination to eligible populations [16,56,57]. Parental hesitancy to vaccinate can be overcome when HCPs engage in educating the parents and patients [58]. A total of five studies, identified that HCP recommendations for vaccination increased the odds of vaccination [16,56,57,58,59]. Moss et al. [58] used NIS-Teen survey data (2010) to determine whether provider recommendation for meningococcal vaccine increased vaccination among adolescents (13–17 years old) compared with adolescents whose parents reported no provider recommendation and found that meningococcal vaccination was higher among adolescents whose parents reported provider recommendations compared with those who received no provider recommendation (77% vs. 54%, *p* < 0.001). Another analysis of the 2010 NIS-Teen survey reported similar results [56]. Vaccination coverage was significantly higher among parents who received a provider recommendation compared with parents who did not receive a provider recommendation: 77.3% vs. 49.7% (*p* < 0.001) [56]. Darden et al. [59] combined 3 years of NIS-Teen data (2008–2010) to identify reasons parents do not vaccinate their adolescents (13–17 years old). Over the 3 years, the most common reason was “Not needed or not necessary.” The proportion of adolescents not up to date (by parent report) decreased from 68.8% in 2008 to 62.6% in 2010. Gargano et al. [57] studied the impact of physician recommendations on parental adolescent immunization attitudes among parents in one county of Georgia (n = 114), including the MenACWY vaccine, and found that among parents who had their adolescent vaccinated, the most common reason reported for their decision to vaccinate was “recommendation by the family practitioner” (91.3%).

An additional study conducted by Healy et al. [60] compared vaccination barriers reported by parents of foreign-born and US-born adolescents (13–17 years old) using NIS-Teen data (2012–2014). While the most common reason for both parents of foreign-born and US-born adolescents for not vaccinating their adolescent with the MenACWY vaccine was “lack of provider’s recommendation” (parents of foreign-born adolescents: 33.9% (95% CI, 27.0–40.7%); parents of US-born adolescents: 37.2% (95% CI, 36.0–38.5%)), no differences in vaccine coverage were noted between the two groups [60]. When the authors calculated the coverage prevalence for ≥one dose of MenACWY vaccine, results were similar for both groups (approximately 80%). The unadjusted MenACWY vaccination coverage prevalence for ≥one dose was 80.6 (95% CI, 77.2–84.0) for foreign-born adolescents and 77.9 (95% CI, 77.3–78.5) for US-born adolescents. Notably, in this study approximately 13% of parents of US-born adolescents who were unvaccinated believed that vaccines were not needed or necessary, thus, highlighting the importance of HCP recommendation [60].

## 4. Discussion

The main implications from this literature review are that awareness and coverage of meningococcal vaccines in adolescents and young adults vary in different population subgroups, indicating lack of health equity across the population, but the data and methods used for these analyses may not fully capture the impact of race on vaccination because they do not account for mediating or moderating effects of the many other variables included in multivariate analyses. Results of this SLR found differences in vaccination coverage over time for the MenACWY and MenB vaccines based on racial/ethnic background, geographic regions, socioeconomic status, insurance status, and physician access, but the differences were not consistent across studies. In particular, routine immunization coverage rates for the MenACWY vaccine were similar or higher for non-Hispanic Black compared with non-Hispanic White persons. However, coverage rates after an outbreak of meningococcal disease in young adults and coverage rates for MenB, which may also be given to those aged 18 years and above, were lower in non-Hispanic Black persons. Coverage of the MenACWY vaccine was also shown to be lower in rural areas than in urban areas, though this has not been shown for the MenB vaccine, for which coverage rates have been shown to be low in both urban and suburban areas. Studies looking at the impact of poverty on coverage of the MenACWY vaccine have shown either no impact or a lower uptake of the second dose in low-income households. For MenB vaccine, coverage has been shown to be higher in high-income areas in one study. Before passage of the ACA, differences in MenACWY vaccine coverage among individuals with private and public insurance were shown in some studies, with higher coverage in those covered by Medicaid, but after passage of the ACA, MenACWY vaccine coverage was similar between those covered by private insurance and those covered by public insurance. For MenB vaccine, individuals with private insurance have been shown to have a higher series completion rate. Finally, access to preventive health care is generally lower in older adolescents and young adults, which is likely to be associated with the lower coverage rates for MenB than for MenACWY vaccines.

The implementation of the ACA requiring health insurance coverage for all ACIP-recommended children’s vaccines (including both MenACWY and MenB vaccines), the Vaccines for Children Program for those without health insurance, and some state requirements for vaccines for school attendance have resulted in high coverage rates for the MenACWY vaccine for all adolescents (e.g., an increase from ~60% [42] to ~80% [43] in MenACWY vaccine coverage). The impact on MenB vaccine coverage is less consistent, as many adolescents wait until age 18 years or older before getting this vaccine and because of uncertainty on the part of providers and parents about the insurance coverage for this vaccine for young adults [12,48,49,53].

As noted by Hogue et al. [61], although the ACIP February 2019 recommendations and guidance introduced new “shared clinical decision-making” (SCDM) for select vaccines, including MenB, this approach is not new to HCPs. Historically, HCPs (including pharmacists) have implemented discussion under SCDM to ensure every patient is fully immunized [61]. However, the SCDM designation by the ACIP, has been interpreted by many HCPs to mean that vaccines with this designation are not covered by third party insurance [61]. Results from this SLR examining US meningococcal vaccination since the publication of ACIP’s recommendations for routine MenACWY vaccination and shared decision-making for MenB vaccination of otherwise healthy adolescents and young adults identified a lack of consistent interpretation and implementation of shared decision-making by HCPs and a lack of parent/guardian awareness of meningococcal disease and the differences in the recommendations for each of the two available types of meningococcal vaccines (MenACWY and MenB). This SLR observed confusion around the interpretation of SCDM and barriers related to financial reimbursement among surveyed HCPs [48,49]. Financial considerations were found to impact the HCP decision to prescribe either of the meningococcal vaccines [49]. Survey results from Kempe et al. [48] identified that HCPs mistakenly believed that vaccines with an ACIP SCDM recommendation were not covered by insurance.

Physician recommendations are a key driver in parents’/guardians’ decision to vaccinate adolescents, with five of the surveys identified by this SLR all noting that meningococcal vaccination was higher among parents of an adolescent who received an HCP recommendation compared with those who had not received a recommendation [56,57,58,59]. These findings are similar to those identified by a Mott poll among parents of adolescents aged 13–17 years which showed the primary way the parents knew when their teen was due for another vaccine was either because the doctor’s office scheduled an appointment for vaccination (44%) or the doctor or nurse mentioned the vaccination while the parent/adolescent was at the office (40%) [62]. A discrete-choice experiment conducted by Johnson et al. [63] found that parents of adolescents and young adults place a significant value on obtaining information pertaining to protection against low-incidence diseases such as meningococcal serogroup B disease that can result in severe long-term disabilities and death. Respondents were willing to pay approximately $400 for 5 years of protection and approximately $100 more to obtain protection against a disease that could result in death and long-term disability [63].

While college students are known to be at increased risk for meningococcal disease, Alderfer et al. [64] note that 18–24 year old noncollege individuals are also at risk for the disease. Alderfer et al. [64] compared incidence of IMD occurring among 18- to 24-year-olds based on college attendance. The authors reported that 64 of 158 cases of IMD (40.5%) occurred in noncollege 18- to 24-year-olds. For MenB vaccination, 0% (none) of the reported cases among noncollege students had received MenB versus 0–7% in those attending college This analysis also, identified disparities in MenACWY vaccination coverage among cases of meningococcal disease in noncollege 18- to 24-year-olds versus those in college, with 90–100% of college cases reporting vaccination with MenACWY versus 38–57% of noncollege cases reporting vaccination with MenACWY. These results highlight that while non-college adolescents bear a significant portion of disease burden meningococcal vaccination is most often discussed for those who are college-bound [64]. Additionally, the most recent CDC meningococcal disease surveillance report (2019) continues to show similar disparities in meningococcal vaccination status in individuals aged 18–24 years attending college versus those not attending college. In 2019, 95.4% of reported cases in this age group had information on college attendance with 51.2% of confirmed cases in students attending college [3]. Of the 21 of 21 college students (100%) with information on MenACWY receipt, 95.2% reported receiving ≥one dose of the MenACWY vaccine. For persons not attending college, 16/20 (80%) had vaccination information, with 75.0% receiving ≥one dose of the MenACWY vaccine. The rates of vaccination for MenB in both groups were lower, with 56.3% of college cases reporting receipt of ≥one dose of the MenB vaccine, and for persons not attending college, of the 55% (11/20 cases) with vaccine information, none had received the MenB vaccine [3]. Outbreaks are potentially preventable through vaccination; however, MenB coverage continues to be less than optimal (range, 9–22%) [51]. Increasing MenACWY and MenB vaccination coverage is necessary to protect all individuals during outbreaks [65,66].

Outbreaks of IMD caused by serogroups A, B, C, W, and Y continue to occur, with an estimated death rate of 0.01/100,000 population in 2019 [22,67,68]. Adolescents and young adults are at increased risk of contracting IMD as a result of typical social behaviors (close living quarters, sharing food and drinks, intimacy, kissing, close contact, smoking, and frequent visits to crowded venues). Importantly, adolescents and young adults are the primary carriers and source of transmission of meningococcal disease [66]. A retrospective review of meningococcal disease outbreaks in the US (2009–2013) conducted by Mbaeyi et al. [68] identified a total of 3686 cases of meningococcal disease. Of the 3686 cases, 4.9% occurred as part of 36 outbreaks (17 organization based and 19 community based). Among the community-based cases, greater than one-third were reported in persons of Hispanic ethnicity (note: result was driven by a small number of outbreaks) [68]. Among university and organization-based outbreaks, the investigators noted that there has been a shift in serogroup distribution from predominately meningococcal serogroup C to serogroup B. This shift was attributed to the success of the routine adolescent MenACWY vaccination program and the less than optimal coverage rates for MenB [68]. The most recent report from National Meningitis Association of Meningococcal Disease on US college campuses, 2013–2019 shows a similar pattern of confirmed cases of meningitis due to serogroup B (Appendix A) [69].

In the US, routine childhood immunization has resulted in sustained reductions in the incidence of vaccine-preventable diseases and remains an effective public health intervention for disease avoidance [70]. Surveillance data from the CDC show a decrease in the incidence of *Haemophilus influenzae* type b (Hib), hepatitis A and B, invasive pneumococcal disease, and varicella following ACIP recommendations for their inclusion in the routine vaccination series for children and individuals 16–23 years old [70]. Between 1980 and 2005 (following publication ACIP recommendations for inclusion of these vaccines as part of routine childhood vaccination programs), diseases such as invasive pneumococcal disease, invasive Hib, hepatitis A and B, and varicella have decreased by more than 80% [70]. From 2017 to 2021, the incidence of diphtheria, Hib, measles, polio, rubella, and tetanus has been reduced to <1 per 100,000 [70]. In order to observe low incidence levels of vaccine-preventable diseases, it is critical to maintain and improve vaccination coverage rates [70]. Similar reductions in the incidence of meningococcal disease in the US have been observed since the late 1990s [1]. Reductions in the meningococcal disease burden have been attributed to the availability of meningococcal vaccines and subsequent ACIP recommendations for their use [17].

### 4.1. Strengths and Limitations

One of the strengths of this study is its use of a comprehensive literature search using Cochrane Equity Methods Group based on the PROGRESS-Plus framework, which refers to a list of characteristics identified by Cochrane that are associated with health equity. Additionally, the review summarized quantitative studies of both vaccine awareness and vaccine coverage and their relationship in different population subgroups. However, there were several limitations identified, particularly study design, as part of the review. A total of ten of the included studies used the CDC NIS-Teen survey dataset showing coverage rates for routinely recommended adolescent vaccines among adolescents 13–17 years of age [71]. Despite the valuable data that are presented, the dataset is limited to young adults aged < 18 years and does not include young adults (18 to 25 years old), as this age group has fewer recommended vaccinations and routine visits. For example, the MenB vaccine is recommended on the basis of shared decision-making for healthy adolescents and young adults aged 16–23 years, with a two-dose series and a preferred age of vaccination between 16 and 18 years, and thus the NIS-Teen survey dataset would not have captured complete coverage rates for this population.

In order to better understand how racism adversely affects health, there are study design and methodologies that should be considered, such as study population and analysis. The role of racism in driving racial health inequities is inexplicit and undertheorized as a clinically relevant cause of racially disparate health outcomes [72]. The majority of the included studies were not found to be generalizable and excluded specific marginalized communities, such as indigenous groups [73,74].

The results from the studies in this SLR that included estimates of the association of race with MenACWY vaccine coverage rates generally did not show a strong correlation. However, a recent review by Williams et al. [75] about the evidence showing the impact of race on health has pointed out some limitations in the available data as well as the analytic challenges in these studies: “These analytic challenges are further exacerbated by difficulties disentangling the potential mediating and moderating effects that contribute to observed patterns. Many studies adjust for variables like poverty or other indications of low socioeconomic status and the social context which are likely a part of the pathway by which segregation exerts its effects. Future research needs to identify the proximal mechanisms linking segregation to health by using longitudinal data to establish temporality and leveraging new statistical techniques.” [75] (p. 109) In this SLR, the multivariate analyses in many of the reviewed studies, using standard statistical techniques, included independent variables such as poverty as well as receipt of other childhood vaccines that are likely to be strongly related to coverage of meningococcal vaccines. But these studies did not present data on the characteristics of individuals who were below the poverty level or who had received previous vaccines, so the impacts of race or other disparities of interest in this review were not clearly estimated. Moreover, non-disaggregated outcome measures limit the ability to understand fully the relationships between ineffective measures of racialization and its impact on health outcomes of Black, Indigenous, and people of color and Hispanic communities [72,76].

### 4.2. Conclusions

Vaccines are the single most important intervention for preventing illness and death in children, adolescents, and young adults. Despite current recommendations for meningococcal vaccination, adolescents and young adults are at increased risk for “under-immunization” with immunizations, including the MenB vaccine [77]. Current evidence demonstrates limitations in study methodologies that likely underestimate the observed health inequities among marginalized communities across factors such as race/ethnicity, socioeconomic status and geography. To reduce societal burdens associated with meningococcal disease, there is an urgent need for future studies to better understand health disparities observed in adolescents and young adults. Clarifications are needed surrounding the difference in ACIP routine recommendation for the MenACWY vaccine and SCDM recommendation for the MenB vaccine for HCPs and parents of adolescents and young adults. In addition, there must be an enhancement of existing strategies aimed at developing disease awareness campaigns and the increased clarity in accessing all meningococcal vaccines nationally.

## Figures and Tables

**Figure 1 vaccines-11-00256-f001:**
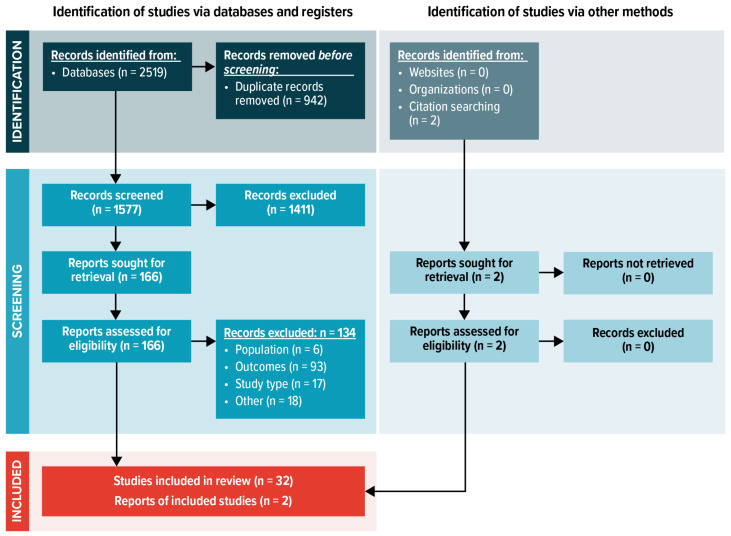
PRISMA Diagram. PRISMA = Preferred Reporting Items for Systematic Reviews and Meta-Analyses.

## Data Availability

Data in this study are presented within the article and Appendix A.

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
