# Peer review of "The Impact of Social Determinants of Health on Meningococcal Vaccination Awareness, Delivery, and Coverage in Adolescents and Young Adults in the United States: A Systematic Review"

_vaccines, 2023, doi:10.3390/vaccines11020256_

Round 1

Reviewer 1 Report

The paper entitled "The Impact of Social Determinants of Health on Meningococcal 3 Vaccination Awareness, Delivery, and Coverage in Adolescents 4 and Young Adults in the United States: A Systematic Review"

The study provides a throughout and well organized literature overview of the drivers of the uptake of MenACWY and MenB vaccines. I recommend the paper for publication.

I have just a very few minor comments:

- I could not find in the paper how the search is done in the different databases. It would be good to report the string in the paper:  "xx" & "yy" & "zzz" & etc.

- It would be good if the paper could name the two persons who did the level 1 and level 2 screening, and the 3rd person who decided in case of controversy.

Reviewer 2 Report

This is a very well-structured study with very few errors to correct for most parts.

Introduction: The topic of interest is comprehensively presented.

However, some clarifications are necessary with regard to the study population: lines 51-52 identify the population of adolescents and young adults with the age range 16-23 years. This contrast with what is stated in the objectives of the study (lines 89 and 97). Please specify this better.

Lines 86-97: The definition of the objective must be unambiguous, therefore, please merge the content of lines 86-89 with that of lines 94-97.

Methods: As guidance, the PICO framework and its variants should be used to formulate the search string: this instrument is essential for the transparency and the reproducibility of the paper (https://pubmed.ncbi.nlm.nih.gov/17653438/).

Formulate the PICO used and report it in the annex.

Among the evidence included in the search are abstracts: why have they been included, as abstracts may not contain adequate information, and the information in abstracts may not be dependable? Please justify the reasons for this choice.

Rows 111-112: the time criterion should be included in the electronic database searches, and not be used in the article screening process.

Results: the results summarise the articles included in the review in a comprehensive manner. Although very long, this part is very interesting and its reading can be facilitated by consulting the tables in the accompanying materials.

Discussion: The paragraph is well structured, although the main implication and further issues do not clearly emerge (the only ones reported are from an article by Williams et al.). These two aspects need to be brought out more.

In addition to the limitations, the strengths of the study also need to be highlighted.
